# Magnetic Nanoparticles with Fe-N and Fe-C Cores and Carbon Shells Synthesized at High Pressures

**DOI:** 10.3390/ma16227063

**Published:** 2023-11-07

**Authors:** Rustem H. Bagramov, Vladimir P. Filonenko, Igor P. Zibrov, Elena A. Skryleva, Boris A. Kulnitskiy, Vladimir D. Blank, Valery N. Khabashesku

**Affiliations:** 1Vereshchagin Institute of High Pressure Physics, Russian Academy of Sciences, Troitsk, Moscow 108840, Russia; filv@hppi.troitsk.ru (V.P.F.); zibrov@hppi.troitsk.ru (I.P.Z.); 2Department of Materials Science of Semiconductors and Dielectrics, National University of Science and Technology MISiS, Moscow 119049, Russia; easkryleva@gmail.com; 3Technological Institute for Superhard and Novel Carbon Materials, Troitsk, Moscow 108840, Russia; boris@tisnum.ru (B.A.K.); vblank@tisnum.ru (V.D.B.); 4Department of Materials Science and Nanoengineering, Rice University, Houston, TX 77005, USA

**Keywords:** iron carbide, iron nitride, carbon core–shell, nanoparticles, high-pressure synthesis

## Abstract

Nanoparticles of iron carbides and nitrides enclosed in graphite shells were obtained at 2 ÷ 8 GPa pressures and temperatures of around 800 °C from ferrocene and ferrocene–melamine mixture. The average core–shell particle size was below 60 nm. The graphite-like shells over the iron nitride cores were built of concentric graphene layers packed in a rhombohedral shape. It was found that at a pressure of 4 GPa and temperature of 800 °C, the stability of the nanoscale phases increases in a Fe_7_C_3_ -> Fe_3_C -> Fe_3_N_1+x_ sequence and at 8 GPa in a Fe_3_C -> Fe_7_C_3_ -> Fe_3_N_1+x_ sequence. At pressures of 2 ÷ 8 GPa and temperatures up to 1600 °C, iron nitride Fe_3_N_1+x_ is more stable than iron carbides. At 8 GPa and 1600 °C, the average particle size of iron nitride increased to 0.5 ÷ 1 μm, while simultaneously formed free carbon particles had the shape of graphite discs with a size of 1 ÷ 2 μm. Structural refinement of the iron nitride using the Rietveld method gave the best result for the space group P6_3_22. The refined composition of the samples obtained from a mixture of ferrocene and melamine at 8 GPa/800 °C corresponded to Fe_3_N_1.208_, and at 8 GPa/1650 °C to Fe_3_N_1.259_. The iron nitride core–shell nanoparticles exhibited magnetic behavior. Specific magnetization at 7.5 kOe of pure Fe_3_N_1.208_ was estimated to be 70 emu/g. Compared to other methods, the high-pressure method allows easy synthesis of the iron nitride cores inside pure carbon shells and control of the particle size. And in general, pressure is a good tool for modifying the phase and chemical composition of the iron-containing cores.

## 1. Introduction

During the last two decades, the nanoparticles of the “core–shell” type have become a hot topic of research due to rapidly expanding areas of their potential high-tech applications [1,2,3]. The cores are physically and chemically active, and the protective shells serve particle stabilization and surface passivation purposes. Graphite carbon shells, for example, counteract the agglomeration of nanoparticles, contribute to their biocompatibility, and are also suitable for functionalization in biomedical applications.

The Fe_3_N@C nanoparticles have already shown excellent electrochemical Li-ion storage properties with high initial charge/discharge capacity [4]. Although the catalytic activity and stability of Fe- and Co-N/C catalysts in oxygen reduction reactions are still below those of Pt-based catalysts, it is believed that under the strong driving force of fuel cell commercialization, Pt-free cathode catalysts with methanol tolerance, such as core–shell Fe- and Co-N/C nanostructures, are particularly attractive candidates to address the cost issue of fuel cell catalysts [5]. Iron carbides and nitrides have a broader spectrum of applications and can be used in biology and medicine, photocatalysis and electrocatalysis, sensorics, and chemistry, as well as magnetic materials and other fields [1,2]. Onion-like carbon nanostructures have great potential for bio-applications [6]

Due to high demand, an active search is currently underway for methods of synthesis of core–shell nanomaterials with pure iron and its carbides. The most commonly used methods are pyrolysis, arc discharge, vapor deposition, as well as detonation technique [7], sonochemical synthesis [8], ball milling [9], plasma synthesis [10], reactions at autogenous pressure at elevated temperatures [11], and others. In comparison, nanoparticles of iron nitrides have been studied much less than carbides.

The high-pressure method for the synthesis of core–shell nanoparticles is only sparsely explored. There are few documented works demonstrating the potential applicability of high pressures for the synthesis of core–shell nanoparticles with iron carbides [12,13,14,15,16]. To the best of our knowledge, core–shell particles with iron nitride cores are investigated much less than iron carbides, although there are published works showing their potential as a catalyst in sustainable zinc–air batteries [17] and as an efficient catalyst for oxygen reduction reactions [18]. There are difficulties that prevent the use of known synthesis methods, including the competition of carbon and nitrogen in reactions with iron, the high volatility of nitrogen at normal pressure, and the particular limitations of the methods.

This work is dedicated to the synthesis of core–shell nanoparticles with iron carbide or nitride cores at high pressures of up to 8 GPa and temperatures around 800 °C. The synthesis of core–shell particles with iron nitride cores and carbon shells is a special task that is difficult to perform using other methods. The high-pressure method is more efficient. There are some interesting aspects to high-pressure synthesis. Under these conditions, the “solid state” [14] diffusion of matter is hindered, and the transformation takes place at the nanoscale. The relative stability of the iron compounds obtained at variable pressures and temperatures was estimated. The structure and chemical composition of the particles were studied using X-ray diffraction, electron microscopy, and X-ray photoelectron spectroscopy. In addition, the magnetic properties of iron nitride core–shell nanoparticles were also measured.

## 2. Materials and Methods

Ferrocene Fe(C_5_H_5_)_2_ (98%, Sigma-Aldrich, St. Louis, MO, USA) and melamine C_3_H_6_N_6_ (99%, Sigma-Aldrich), selected as starting materials, were mixed (70/30% by weight) in benzene medium in an agate mortar using ultrasound and then air-dried at 50 °C. The resulting mixture is henceforth designated as FerrMelam.

High-pressure experiments were carried out in a “toroidal” apparatus [19]. The sample was loaded into a graphite ampoule, which also acted as an electrical heater, and then placed in a container made of lithographic stone. When compressed by profiled tungsten carbide anvils, this container experienced plastic flow and transmitted pressure to the working area. The pressure was calibrated using the generally accepted method of recording the phase transitions of reference substances by measuring their electrical resistance [20]. Temperature was measured with thermocouples.

The samples discussed in this paper were obtained using a similar procedure. That is, the starting ferrocene or FerrMelam was first pressurized to the desired pressure, then heated to 800 °C for 600 s, unless otherwise stated, and held at that temperature for 30 min. Several samples were treated at temperatures above 800 °C. The starting FerrMelam was first loaded to 8 GPa, heated to 800 °C for 10 min, held for 0.5 min, then heated to 1200 or 1600 °C for 0.5 min, and finally held for 0.3 min.

The microstructure was studied using a JEM-2010 (JEOL, Akishima, Tokyo, Japan) transmission electron microscope with an energy-dispersive X-ray spectroscopy (EDS) attachment.

The G670 imaging plate Guinier camera (HUBER SE, Berching, Germany) was used to perform X-ray phase and structure analysis, using Cu Kα1 radiation.

A PHI 5500 ESCA (Physical Electronics, Inc. (PHI), Chanhassen, MN, USA) spectrometer with Al Kα radiation (hν = 1486.6 eV, 200 W) was used to conduct X-ray photoelectron spectroscopy (XPS). The high-resolution spectra had a resolution better than 0.65 eV. Differences in surface and volume were evaluated through surface etching using Ar+ ions.

The magnetic properties of the synthesized samples were measured using a LakeShore Cryotronics-7404 magnetometer (LakeShore Cryotronics, Inc., Westerville, OH, USA) at room temperature under magnetic fields of up to 7.5 kOe.

## 3. Results

Previous research studies [12,13,14,15] have shown that the transformation of ferrocene starts with its decomposition at 500 ÷ 600 °C under pressures of 2–8 GPa. Under these pressure/temperature (P/T) conditions, an amorphous, disordered product with iron-atom-enriched regions is formed [15]. Increasing temperature and/or holding time results in the formation of iron carbide nanoparticles surrounded by graphite shells. The transformation process is relatively slow due to the primary limiting factor being diffusion at low temperatures. For example, at a temperature of 800 °C, it takes tens of minutes to form stable core–shell nanoparticles [12].

In this work, electron microscopy analysis showed that thermobaric treatment of ferrocene or a mixture of ferrocene/melamine results in the formation of a large number of nanoscale core–shell particles throughout the studied pressure range of 2 ÷ 8 GPa. EDS (energy-dispersive X-ray spectroscopy) analysis revealed that the cores are iron compounds surrounded by carbon shells. This transformation requires a temperature of about 800 °C and an exposure time of tens of minutes. The average size of the particles obtained at 800 °C did not exceed 60 nm (Figure 1 and Appendix A).

X-ray phase analysis (Figure 2) showed that iron carbide Fe_3_C is formed from ferrocene at 800 °C and pressures of 2 and 4 GPa, while iron carbide Fe_7_C_3_ is formed at 8 GPa. TEM images of single Fe_3_C@C and Fe_7_C_3_@C nanoparticles are shown in Figure 3a,b. The insets show the corresponding FFT images. Note that several different crystal lattices can exist for the Fe_7_C_3_ composition [21]. In particular, the lattice shown in Figure 3b has a hexagonal structure with parameters: a = 1.398 nm, c = 0.4506 nm.

The atomic Fe/C ratio in Fe_3_C carbide is 3, and for Fe_7_C_3_ it is 2.33. Thus, the data obtained show that when the synthesis pressure is increased from 2 ÷ 4 to 8 GPa and the synthesis temperature is below 800 °C, the atomic Fe/C ratio in the carbide decreases from 3 to 2.33. However, when the experiment is carried out at 8 GPa and a temperature of up to 1600 °C, the formation of Fe_3_C is observed again, in agreement with the literature data [13].

On the contrary, HPHT treatment of the FerMelam mixture under similar conditions (800 °C/30 min) resulted in the formation of iron nitride cores only at all pressures in the range of 2 ÷ 8 GPa (Figure 4). It should be noted that the atomic content of carbon is more than three times higher than the atomic content of nitrogen in the initial mixture. This suggests that at pressures of 2 ÷ 8 GPa the obtained iron nitrides are thermodynamically more stable than iron carbides.

It was found that the formation of core–shell nanoparticles from FerMelam at 800 °C was significantly dependent on the exposure time. A 30-min exposure time was found to be sufficient for the formation of such particles with dimensions of 40 ÷ 60 nm (Appendix A). It was also noted that at a similar temperature, the higher the pressure, the less perfect the shells around the core. TEM analysis of the carbon matrix of the samples synthesized at 800 °C and below showed that it contains some flat and some bent graphite nanopacks, and also areas of amorphous carbon. Its content varies and depends on the synthesis conditions. However, a detailed analysis of the carbon matrix is beyond the scope of this article.

TEM analysis of the layered carbon shells revealed that they can contain 5–50 layers which are flat in some places and curved in others. Figure 5 shows a particle obtained from FerMelam at 4 GPa/800 °C/30 min. The curved sections of the shell are usually more defective. A high-resolution image of the shell is shown in Figure 5b. A contrast formed by the (002) graphite planes and some planes intersecting them at 70° to 80° angles can be observed. The interlayer distance of these intersecting planes is about 0.2 nm. This shows that the packing sequence of the (002) graphite layers forms complex polytypes that are not common ABAB ones. Both of the layer systems are shown in the insets in Figure 5b, which are marked with white frames. The corresponding Fourier transform image is shown in the lower right corner inset, where the brighter reflection corresponds to (002) graphite planes, and the less bright one marked by an arrow is associated with intersecting layers. Thus, these data show that the graphite-like shells can sometimes be packed in a rhombohedral shape.

In the samples obtained at temperatures and/or exposure time insufficient for the formation of stable shells, in addition to iron nitride particles (Figure 5), particles containing iron oxide with a tetragonal lattice were occasionally found (Figure 6). In the surface layers of iron, which acted as a catalyst for the growth of carbon nanostructures, oxygen was previously observed [22], and iron oxide Fe_2_O_3_ was detected. The presence of another iron oxide, Fe_3_O_4_, in the surface layers of core–shell structures was also previously observed [16]. In our case, a rare tetragonal phase of Fe_2_O_3_ composition was found inside the carbon shells. The crystal structure of the iron oxide γ-Fe_2_O_3_ is usually reported either in the cubic system (space group P4_3_32) with partial Fe vacancy disorder or in the tetragonal system with full site ordering and c/a ratio ~ 3 [23]. A tetragonal lattice differs from a cubic one by a triple value of one parameter. Thus, the maghemite structure has the tetragonal space group P4_1_2_1_2 with a = 0.8347 nm and c = 2.5042 nm (spinel cubic cell parameter tripled along the c-axis). It has been shown [20] that the tetragonal structure of γ-Fe_2_O_3_ is more stable than the cubic one. The tetragonal phase of Fe_2_O_3_ has been previously obtained only in small amounts in the preparation of magnetically active nanocomposite aerogels [24] and in the CVD process [25].

The change in the structure of the material obtained from the FerMelam mixture at temperatures above 800 °C can be tracked by the experimental data presented below, as follows: The samples were first loaded to 8 GPa, heated to 800 °C for 10 min, held for 0.5 min, then heated to 1200 or 1600 °C for 0.5 min, and finally held for 0.3 min. The samples were then cooled for 10 s. The X-ray diffraction patterns of the material obtained are shown in Figure 7a. All lines can be attributed to the graphite phase and to iron nitride Fe_3_N_1+x_. The general appearance of the diffraction patterns indicates that with increasing temperature the lines of these phases become narrower, and their intensity increases. This indicates the ordering of the structure and the reduction in defects, both in the carbon matrix and in the nitride. According to the results of scanning and transmission electron microscopy, the free carbon in the 1600 °C sample consisted of disc-shaped graphite particles with a diameter of about 1 ÷ 2 μm. The size of the round iron nitride particles was 0.2 ÷ 1.5 μm (Figure 7b).

Full profile analysis of the X-ray diffraction data (Rietveld refinement) was used to refine the structure of iron nitrides obtained from FerMelam mixtures at 8 GPa/800 °C/30 min and 8 GPa/800 °C/0.5 min + 8 GPa/1600 °C/0.3 min. Three different nitride space group models were tested: P 312 [26], P-31m [27], and P 6322 [28]. The following structural reliability factors *R_F_* were obtained for the sample (8 GPa/800 °C/600 s + 8 GPa/1600 °C/20 s): P 312 − *R_F_
*= 0.0913; P-31m − *R_F_
*= 0.1000; P 6322 − *R_F_
*= 0.0757. Thus, the best refinement parameters were obtained in the *P 6322* group; the results are shown in Figure 8 and in Table 1 and Table 2. Refined chemical composition for FerMelam/8 GPa/800 °C/30 min is Fe3N_1.208_ and for FerMelam/(8 GPa/800 °C/0.5 min + 8 GPa/1600 °C/0.3 min) it is Fe_3_N_1.259_.

XPS studies were carried out to clarify some details of the atomic composition and chemical bonding of the raw and ion-etched surface of the sample obtained from the ferrocene–melamine mixture via treatment at 4 GPa/800 °C/30 min. The survey spectra are presented in Appendix A. XPS high-resolution C1s, N1s, and Fe2p peaks are shown in Figure 9. All tests indicated that carbon is the main component (97 at%) of the sample surface. Its C1s peak can be found at a binding energy position of approximately 284.5 eV (Figure 9a) with a FWHM of approximately 1.0 eV. The asymmetry of the peak and the presence of π-π* satellite show that the carbon has a graphite-like structure.

The XPS tests showed no iron on the raw surface while the oxygen content was above 3 at%. The nitrogen content remained constant before and after surface etching. Tests on the ion-etched surface showed 0.5–1.5 at%, 0.5–1.0 at%, and 0.7–1.2 at% of O, Fe, and N, respectively. As XPS is a surface-sensitive method, it can be concluded that the oxygen is preferentially adsorbed on the surface of the sample; the iron [29] is somehow “hidden” before the surface is etched.

For the N1s peak of nitrogen, the signal-to-noise ratio was low (Figure 9). However, two peaks could be accurately fitted: a narrow one at 398.0 eV (FWHM—1.1 eV) and a broad one at 400.0 eV (FWHM—2.4 eV). These peaks indicate nitrogen atoms inserted into the hexagonal carbon network, which have two atomic neighbors (first peak) and three atomic neighbors (second peak) [30].

The iron Fe2p3/2 peak (approximately 707.5 eV) in Figure 9c can be attributed to the Fe_x_N (2 < x < 4) compound, according to [31]. If this is the case, the N1s peak should be located at 396.9–398.1 eV, and this is what we observed in our data.

Thus, the XPS results evidence that the iron is chemically bound to nitrogen, the carbon is in a graphite-like form and the total nitrogen content in the carbon matrix is about 0.7 ÷ 1.2 at%.

## 4. Discussion

The experiments showed that only iron nitrides are formed from a mixture of ferrocene and melamine in the whole selected range of pressures (2 ÷ 8 GPa) and temperatures (up to ~1600 °C). Considering that the atomic carbon content in the initial 70 wt%Fe(C_5_H_5_)_2_ + 30 wt%C_3_H_6_N_6_ mixture is more than three times higher than the nitrogen content (C/N > 3), this indicates that under the investigated conditions iron nitrides are more stable than iron carbides in core–shell nanoparticles.

The process of formation of core–shell nanoparticles has common features both in the case of iron carbides from ferrocene ([12,15] and present work) and in the case of nitrides from ferrocene/melamine mixture in the present work. At high pressures (2 ÷ 8 GPa), the decomposition of the initial hydrocarbons begins at temperatures of 500 ÷ 600 °C, when the process of concentration (segregation) of iron atoms occurs along with other processes. Unlike the Fe-C system in the case of ferrocene alone, in the Fe-C-N system iron is concentrated in the form of hydride–nitride. This is followed by the formation at higher temperatures of an ordered (crystalline, locally more perfect) nitride, which, with increasing temperature and/or time, approaches a stoichiometric ratio close to Fe_3_N. At 800 °C, this transformation is completed in tens of minutes. In this case, the iron nitride acquires a hexagonal structure with the space group P6_3_22.

It should be noted that high pressure is a factor that reduces the mobility of atoms and the diffusion rate. Therefore, it is believed [14] that the processes discussed here occur in the solid phase at temperatures around 800 °C. Nevertheless, diffusion fluxes at the nanoscale are sufficient to form nanoparticles with an iron nitride core and graphite-like shells.

It was found that the approximate composition of the iron nitride core particles Fe_3_N_1+x_ is preserved when the temperature rises above 800 °C, but at the same time, the particles increase in size up to an average of 0.2 ÷ 1.5 μm at 1600 °C and 8 GPa (Figure 7b). Neither Fe nor N accumulate in the carbon matrix. This conclusion is based on the results of X-ray photoelectron spectroscopy studies. In particular, although a significant amount of nitrogen is present in the initial mixture of 70 wt%Fe(C_5_H_5_)_2_ + 30 wt%C_3_H_6_N_6_ (atomic ratio N/C ~ 0.32), its concentration in the final carbon matrix is significantly lower. An estimate for the sample FerMelam/4 GPa/800 °C/30 min gives a value of 0.7 ÷ 1.2 at% N. At the same time, the chemical bonding of the residual nitrogen atoms in the carbon graphite matrix is the same as in pyridine, meaning that it could be inherited from the original melamine molecules.

Modern views on the equilibrium phases in the Fe-N system can be found in [32,33,34]. The gamma phase γ’ (which can be represented as a solid solution of nitrogen atoms in tetrahedral voids of the FCC lattice of iron) has a narrow range of homogeneity corresponding to the composition of Fe_4_N (N = 0.20 atoms). The stroke in the gamma phase designation means that nitrogen atoms in Fe4N occupy an ordered position. The epsilon ε-phase of iron nitride with an approximate composition of Fe_3_N has a wide range of homogeneity from 15 to at least 33 at%. Increasing the nitrogen content by more than 20 at% stabilizes this nitride with a hexagonal lattice. It is also known that non-nanoscale cubic phases of iron nitride are less stable than hexagonal ones at elevated pressure and temperatures [35,36,37,38].

From the results presented above it can be concluded that the thermodynamic stability of nanoscale phases increases at 4 GPa in the Fe_7_C_3_ -> Fe_3_C -> Fe_3_N_1+x_ series and at 8 GPa in the Fe_3_C -> Fe_7_C_3_ -> Fe_3_N_1+x_ series. Nevertheless, it should be noted that for evaluation of the relative stability of these cores in the core–shell nanoparticles at ambient pressure, a more focused investigation has to be carried out which was not within the scope of the present work.

Our observation is valid for temperatures up to 800 °C. Moreover, at pressures of 2 ÷ 8 GPa, the Fe_3_N_1+x_ iron nitride is more stable than any of the iron carbides, even at temperatures up to 1600 °C. This is in agreement with experimental data on iron nitrides found as inclusions in natural mined diamonds [39]. It is noted that the close structural similarity between Fe-carbide and Fe-nitride indicates that they can form solid solutions at high pressures. Fe-nitride readily replaces Fe-carbide during the reaction of N-bearing fluid with Fe_3_C at 6 ÷ 20 GPa [40]. This may indicate that Fe-nitride is more stable than carbide at high pressures [41]. Our data obtained for core–shell iron nitride nanoparticles synthesized in the presence of hydrogen do not contradict this view.

Previous studies have shown that Fe_7_C_3_@C nanoparticles synthesized at high pressure are magnetic, and testing of their biocompatibility with living cells demonstrated negative cytotoxicity [14]. The results of the magnetic property measurements are important in assessing the controllability of the particles by the magnetic fields. Magnetic measurements of the sample produced from FerMelam at 4 GPa/800 °C/30 min showed that the coercive force is about 42 Oe, and the specific magnetization at 7.5 kOe is 23 emu/g according to the M vs. H curve (Figure 10).

To attribute this result to pure Fe_3_N_X_, its quantity in the sample needs to be estimated. A phase composition was derived from the calculation by assuming that all carbon in the original 70/30% ferrocene/melamine mixture transforms into graphite, all ferrocene iron converts to Fe_3_N, and some nitrogen exits the system. The calculation revealed 29.7 wt% content for Fe_3_N and 70.3 wt% for graphite. Estimates of the phase composition for the sample 8 GPa/800 °C/0.5 min + 8 GPa/1600 °C/0.3 min, using Rietveld refinement, indicate that it consists of 31.79 ± 0.06 wt% Fe_3_N_1.259_ and 68.21 ± 0.08 wt% graphite carbon. Thus, the specific magnetization at 7.5 kOe of the Fe_3_N_1.208_ component of the FerMelam at 4 GPa/800 °C/30 min sample can be estimated as 69 ± 7 emu/g. Although this value does not set a record [42], it still indicates that iron nitride core–shell nanomaterials may be as attractive for practical applications [43] as previously studied iron carbide core–shell nanostructures [14].

## 5. Conclusions

Core–shell nanoparticles with Fe_3_C carbide cores were obtained from ferrocene at 2 ÷ 4 GPa, and with Fe_7_C_3_ carbide cores at 8 GPa. Core–shell nanoparticles with iron nitride cores and stoichiometry Fe_3_N_1+x_ (x = 0.2 ÷ 0.3) were obtained from ferrocene/melamine mixture (7/3 by weight) at 2 ÷ 8 GPa and temperatures approximately 800 °C. The required time for the formation of relatively stable particles is in the range of tens of minutes. The average particle size does not exceed 60 nm. Fe and N atoms do not accumulate in the carbon matrix. This was confirmed by the results of X-ray photoelectron spectroscopy investigations.

If the temperature does not exceed 800 °C, the thermodynamic stability of the nanoscale phases increases at 4 GPa in the Fe_7_C_3_ -> Fe_3_C -> Fe_3_N_1+x_ series and at 8 GPa in the Fe_3_C -> Fe_7_C_3_ -> Fe_3_N_1+x_ series.

Furthermore, the iron nitride Fe_3_N_1+x_ is more stable than any of the iron carbides under all the conditions considered, i.e., pressures of 2 ÷ 8 GPa and temperatures up to 1600 °C. High-pressure–high-temperature treatment of a ferrocene/melamine mixture at 1600 °C and 8 GPa yielded iron nitride Fe_3_N_1+x_ particles with sizes up to 0.2–1.5 μm and free carbon consisting of rounded flat graphite particles with a diameter of about 1–2 μm.

When the structure of iron nitride was refined using the Rietveld method, the best result was obtained for the space group P6_3_22. The refined chemical composition for the sample obtained from the mixture of the ferrocene and melamine (70/30% by weight) obtained at 8 GPa/800 °C/30 min was Fe_3_N_1.208_, and for the sample obtained at (8 GPa/800 °C/0.5 min + 8 GPa/1600 °C/0.3 min)—Fe_3_N_1.259_.

At temperatures of about 800 °C and pressures of 2 ÷ 8 GPa, all transformations take place in the “solid phase” in the absence of the “classical” gas phase. In the Fe-C-N system, the first stage is the decomposition of hydrocarbons, followed by the segregation of iron and nitrogen atoms, and the final stage is the nucleation and growth of core–shell nanoparticles with an iron nitride core at sufficient temperature and time.

The demonstrated magnetic behavior of iron nitride core–shell nanoparticles obtained from the ferrocene/melamine mixture at 4 GPa/800 °C/30 min creates the opportunity for their applications in biology and medicine, and also for the design of reusable catalysts capable of manipulation using magnets in the reaction media [44,45].

Thus, it has been shown that high pressures (up to 8 GPa) can provide an effective tool for the synthesis of core–shell nanoparticles, allowing control of the phase composition of cores consisting of iron compounds such as carbides and nitrides.

## Figures and Tables

**Figure 1 materials-16-07063-f001:**
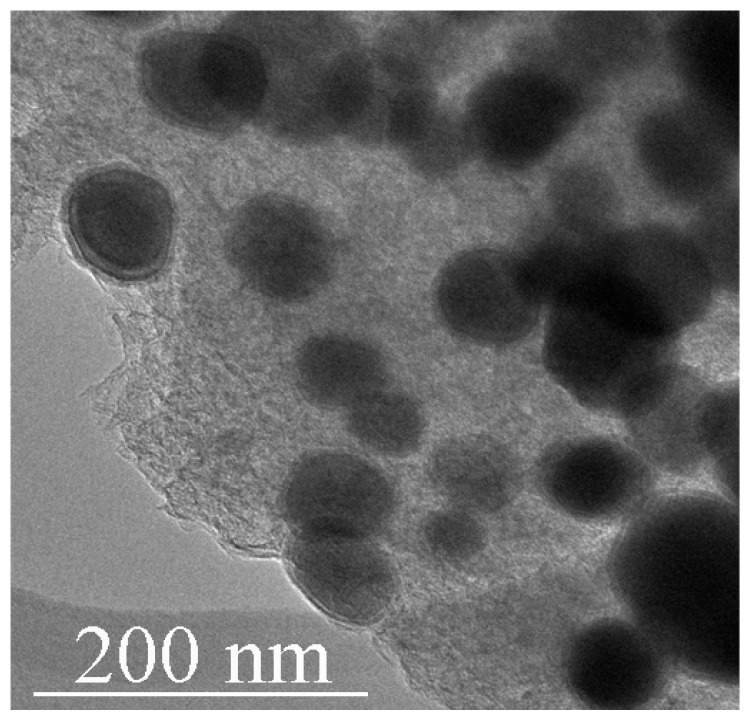
TEM image of Fe_3_C iron carbide particles encapsulated in carbon shells obtained from ferrocene at 2 GPa/800 °C/30 min.

**Figure 2 materials-16-07063-f002:**
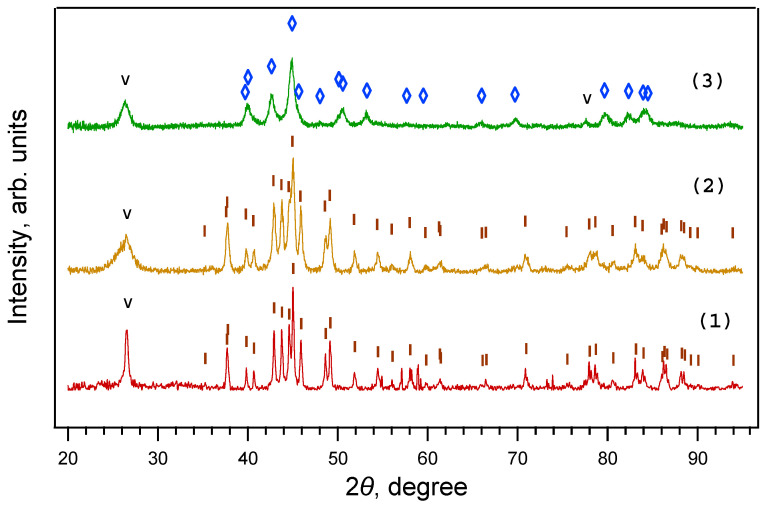
X-ray diffraction patterns of samples obtained from ferrocene under a 30 min exposure at 800 °C and pressures of 2 GPa (1), 4 GPa (2), and 8 GPa (3). Fe_3_C is present only at 2 ÷ 4 GPa, while Fe_7_C_3_ is present only at 8 GPa. ◊—Fe_7_C_3_. I—Fe_3_C; V—graphite.

**Figure 3 materials-16-07063-f003:**
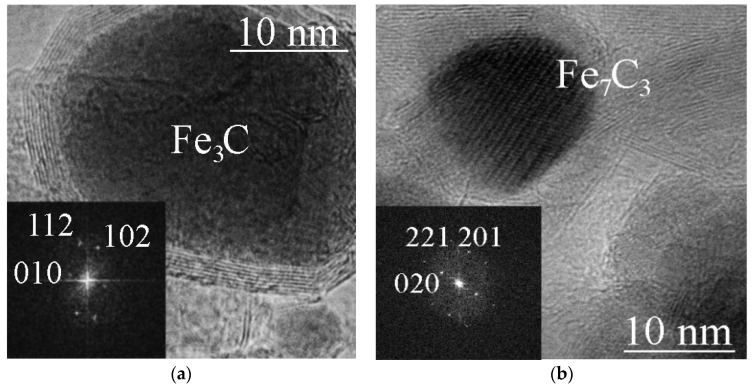
TEM images of a Fe_3_C@C particle (ferrocene/4 GPa/800 °C/30 min) (**a**), and a Fe_7_C_3_@C particle (ferrocene/8 GPa/800 °C/30 min) (**b**). The inserts show FFT images of Fe_3_C cementite (**a**) and Fe_7_C_3_ with a hexagonal lattice (**b**).

**Figure 4 materials-16-07063-f004:**
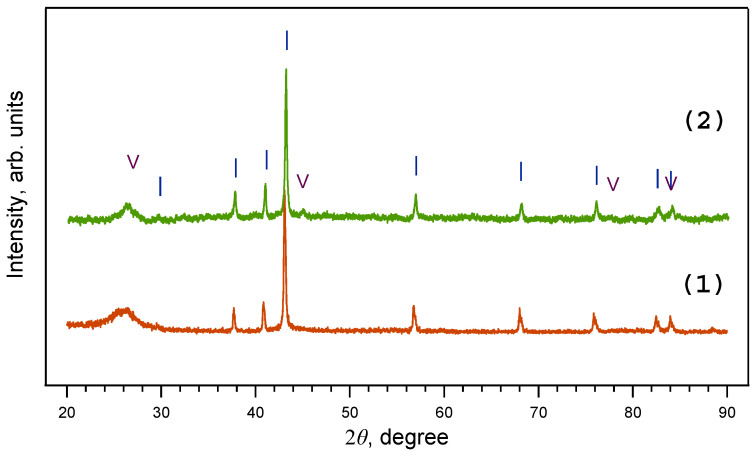
X-ray diffraction patterns of samples obtained from ferrocene/melamine mixture at 2 GPa/800 °C/30 min (1) and 8 GPa/800 °C/30 min (2). Note that at 2–8 GPa only iron nitride Fe_3_N_1+x_ is present. I—Fe_3_N_1+x_; V—graphite.

**Figure 5 materials-16-07063-f005:**
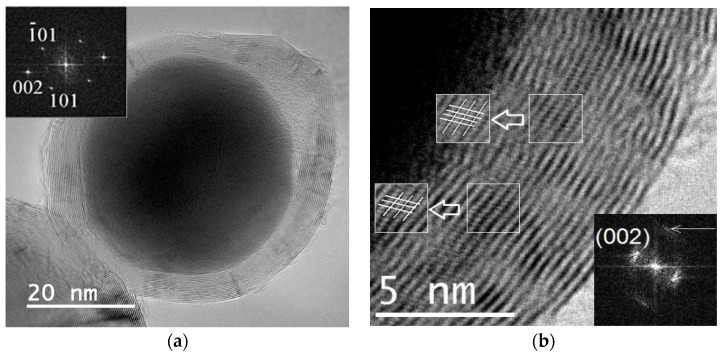
TEM image of a single particle obtained from FerMelam at 4 GPa/800 °C/30 min. Core–shell nanoparticle with Fe_3_N_1+x_ core (**a**). High-resolution image of the carbon shell (**b**).

**Figure 6 materials-16-07063-f006:**
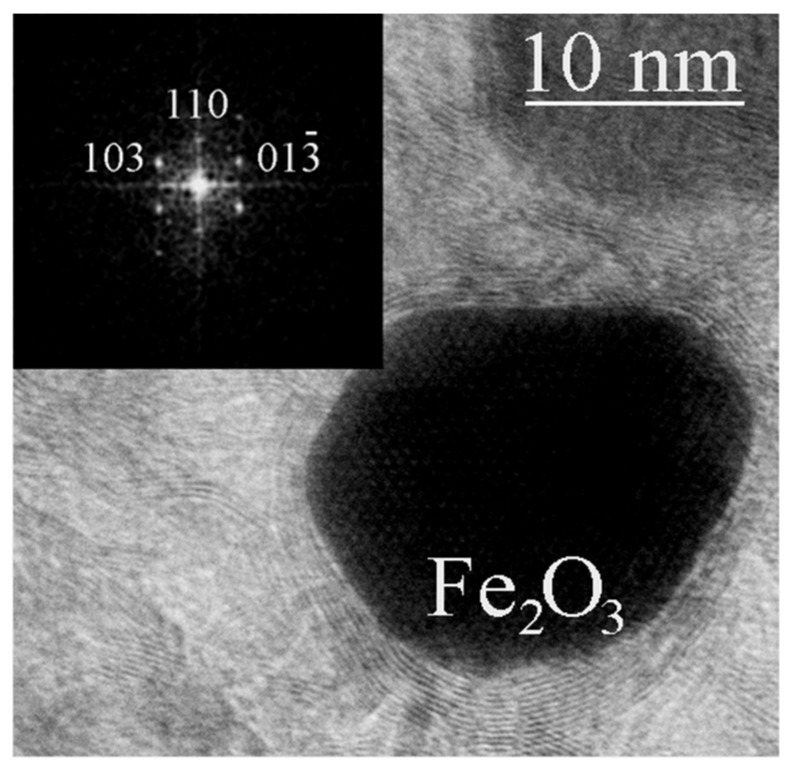
TEM image of a tetragonal γ-Fe_2_O_3_ particle detected in the material obtained from FerMelam at 8 GPa/800 °C/0.3 min. The inset shows the corresponding FFT image.

**Figure 7 materials-16-07063-f007:**
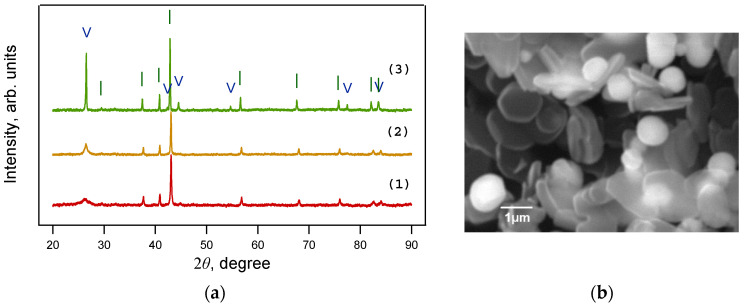
(**a**) X-ray diffraction patterns of samples obtained from the FerMelam mixture at 8 GPa and 800 °C (1); 1200 °C (2); 1600 °C (3). Only Fe_3_N_1+x_ is present at 8 GPa and 800 ÷ 1600 °C. (**b**) SEM image of the sample obtained from the FerMelam mixture at 8 GPa and 1600 °C. I—Fe_3_N_1+x_; V—graphite.

**Figure 8 materials-16-07063-f008:**
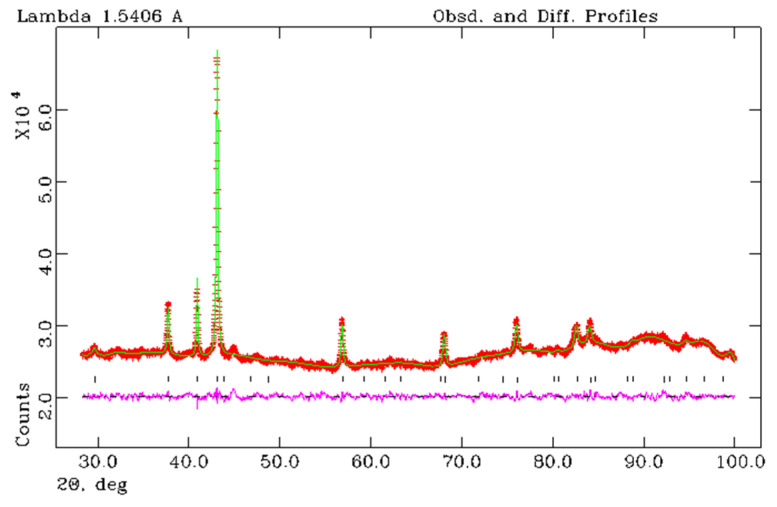
X-ray Rietveld refinement of the sample Fe_3_N_1.208_ obtained from the FerMelam mixture at 8 GPa/800 °C/30 min. The observed (+), calculated (solid line), and the difference between observed and calculated (bottom curve) powder diffraction profiles. Allowed Bragg reflections are indicated by vertical tick marks.

**Figure 9 materials-16-07063-f009:**
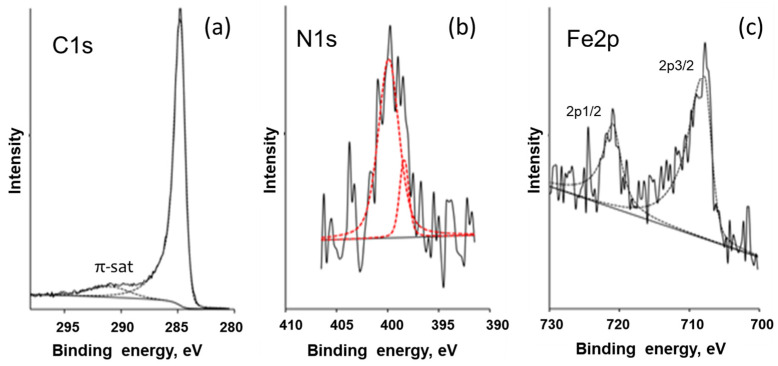
XPS high-resolution C1s (**a**), N1s (**b**), and Fe2p (**c**) peaks shown by solid and dashed (fitted) curves for the sample obtained from FerMelam at 4 GPa/800 °C/30 min.

**Figure 10 materials-16-07063-f010:**
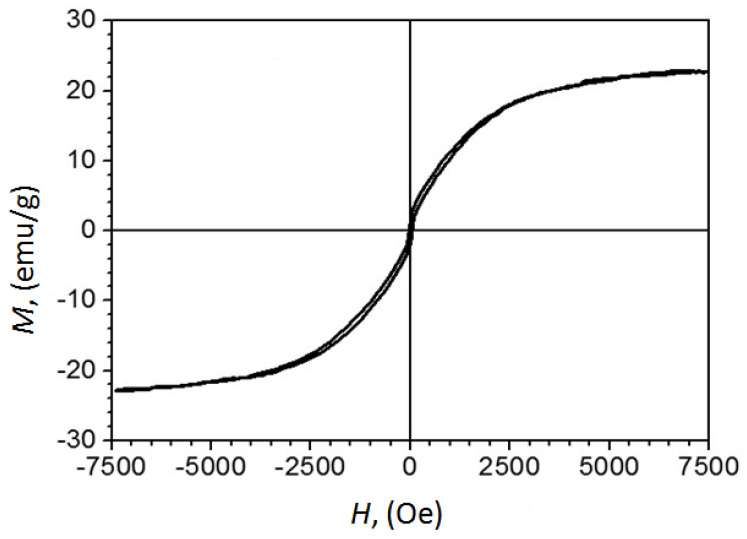
The curve of magnetization versus the applied field obtained at room temperature for the as-prepared Fe_3_N_1.208_@C sample with calculated iron nitride nanoparticle content of ~30% by mass, and carbon shells and matrix of ~70% by mass.

**Table 1 materials-16-07063-t001:** Results of the refinement of the structure using the Rietveld method.

Sample	FerrMelam 8 GPa/800 °C/30 min	FerrMelam 8 GPa/800 °C/0.5 min + 8 GPa/1600 °C/0.3 min
Refined chemical composition	Fe_3_N_1.208_	Fe_3_N_1.259_
Molecular weight	184.46	185.306
Space group	P6_3_22	
*a* (Å)	4.76180 (4)	4.79120 (2)
*c* (Å)	4.40300 (6)	4.41178 (3)
V (Å^3^)	86.461 (1)	87.707 (1)
Z	2	
*d*-calcul. (g/cm^3^)	7.085	7.017
X-ray data collection		
Imaging Plate Guinier Camera G670 (Huber)		
Radiation	CuKα_1_	
Wavelength (Å)	1.5405981	
Temperature (K)	293	
Refinement	GSAS	
*R_F_*	0.1032	0.0757
*R_P_*	0.0083	0.0099
*R_WP_*	0.0105	0.0144

**Table 2 materials-16-07063-t002:** Atomic coordinates, position occupancy, and isotropic thermal parameters U_iso_ (Å^2^) for the FerrMelam/8 GPa/800 °C/30 min sample (Fe_3_N_1.208_).

Atom	Site	OCC	*x*	*y*	*z*	U_iso_
Fe	(6g)	1.0	0.3340 (21)	0	0	0.0198 (3)
N1	(2b)	0.208 (17)	0	0	0.25	0.025
N2	(2c)	1.0	0.3333	0.6667	0.25	0.025

## Data Availability

Data are contained within the article and Appendix A.

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
