# Peer review of "Magnetic Nanoparticles with Fe-N and Fe-C Cores and Carbon Shells Synthesized at High Pressures"

_materials, 2023, doi:10.3390/ma16227063_

Round 1
Reviewer 1 Report
Comments and Suggestions for Authors
(1) The study delves into the formation of iron nitride and iron carbide core-shell nanoparticles under specific conditions. The potential relevance of these findings to broader scientific or industrial contexts should be highlighted. Why is this work significant?
(2) In the introduction, I recommend expanding upon the existing literature by integrating a more comprehensive review of previous studies related to both "core-shell" type nanoparticles and their synthesis methods. This would provide readers with a clearer context and highlight the novelty and significance of your study within the broader research landscape.
(3) While the Rietveld method was used for structural refinement and the results aligned with one space group, it might be helpful to consider or discuss any other external validation or confirmatory tests.
(4) While the manuscript provides detailed experimental results, it doesn't sufficiently connect these findings to broader real-world applications or clarify their significance. For example, it briefly mentions potential uses in biology, medicine, and as reusable catalysts, but there's no comprehensive exploration of why these applications might benefit from the findings or how they could be innovatively employed.
(5) The work highlights the use of high pressures as an effective synthesis tool for nanoparticles but falls short in comparing this methodology with existing or alternative nanoparticle synthesis techniques. A comparative analysis would help position the significance of their method within the broader scientific community and potentially highlight its advantages or limitations.
Author Response
Dear reviewer:
Attached please find the answers to comments you made

Reviewer 2 Report
Comments and Suggestions for Authors
The manuscript by Rustem H. Bagramov and coauthors reports their work on the synthesis and characterization of core-shell structures of Fe-C/Fe-N carbon shell. I have the following concerns discouraging me from recommending publication:
1, the figures of Figures 5 a/b seem misrepresented.
2, the synthesis parameters for the samples should be organized more clearly, such as in a table, to highlight the differences in preparing the samples.
3, how about the results of other refinements based on P312/P-31m? the authors claimed they are worse, how worse is it?
4, I could not see any concrete evidence relating to the higher stability of iron nitride than iron carbide in the core-shell nanoparticles, the authors should point out the evidence clearly if they claim so.
5, the composition of the materials evaluated from XPS or Rietveld refinement on the powder X-ray diffraction is rather unreliable, as the authors pointed out: XPS is a technique for surface analysis, which is problematic to determine the composition of a bulky sample; while Rietveld refinement is a mathematic program to determine the match between diffraction profile and atom arrangement in the crystal lattice, it has very weak connection to the materials exact composition in my opinion. I think elemental analysis to determine the composition of their materials is necessary.
Author Response
Dear Reviewer:
Attached please find the responses to reviewer comments

Round 2
Reviewer 2 Report
Comments and Suggestions for Authors
The authors clarified my concerns. I support publication.